# Sex-Specific Patterns of Cortisol Fluctuation, Stress, and Academic Success in Quarantined Foreign Medical Students During the COVID-19 Lockdown

**DOI:** 10.3390/life16010054

**Published:** 2025-12-30

**Authors:** Vedrana Ivić, Irena Labak, Oksana Shevchuk, Rudolf Scitovski, Viktoria Ivankiv, Kateryna Kozak, Mykhaylo Korda, Marija Heffer, Sandor G. Vari

**Affiliations:** 1Department for Medical Biology and Genetics, Faculty of Medicine Osijek, Josip Juraj Strossmayer University of Osijek, HR-31000 Osijek, Croatia; vedrana.ivic@mefos.hr (V.I.);; 2Department of Biology, Josip Juraj Strossmayer University of Osijek, HR-31000 Osijek, Croatia; 3Pharmacology and Clinical Pharmacology Department, I. Horbachevsky Ternopil National Medical University, 46000 Ternopil, Ukraine; 4School of Applied Mathematics and Informatics, Josip Juraj Strossmayer University of Osijek, HR-31000 Osijek, Croatia; 5Department of Pediatrics No. 2, I. Horbachevsky Ternopil National Medical University, 46000 Ternopil, Ukraine; 6Medical Biochemistry Department, I. Horbachevsky Ternopil National Medical University, 46000 Ternopil, Ukraine; 7International Research and Innovation in Medicine Program, Cedars–Sinai Medical Center, Los Angeles, CA 90048, USA

**Keywords:** SARS-CoV pandemic, lockdown, salivary cortisol, Cohen’s perceived stress scale, medical students, sex difference, mathematical clustering

## Abstract

Cortisol is built into the circadian clock mechanism, but it is also the body’s natural response to stress. Insight into sex-specific cortisol fluctuations may elucidate individual differences in physiological and pathological patterns. This cross-sectional study examined sex-specific adaptation to stress induced by COVID-19 pandemic and lockdown in foreign medical students at I. Horbachevsky Ternopil National Medical University, Ukraine (TNMU). Salivary cortisol was analyzed using cluster-based mathematical modeling to identify natural groupings in the data. Perceived stress was measured using Perceived stress scale-10 (PSS-10). The academic success was accessed from the official records of the TNMU. Average value of area under the curve (AUC) of daily salivary cortisol from the whole sample showed that men had higher cortisol than women. Mathematical clustering explained shift of the cortisol peak, and divided sample into 5 clusters—two of which had predicted daily cortisol pattern and represented most participants (65.6% men and 73.6% women), while the rest had aberrant daily cortisol pattern. Females had higher total PSS-10 score than males. PSS-10 subscales correlated with aberrant daily cortisol pattern. Unexpectedly, COVID-related circumstances did not have impact on participants’ academic success.

## 1. Introduction

When faced with a specific stressor, the organism reacts to adapt and overcome the challenge. This process triggers the activation of the hypothalamic-pituitary-adrenal (HPA) axis, resulting in the secretion of cortisol, the primary stress hormone. In stressful situations, cortisol prepares the body for a response by enhancing metabolism and cognition while inhibiting general vegetative functions, such as digestion, growth, reproduction, and immunity [1]. Typically, cortisol works in synergy with other hormones to maintain homeostasis, and its secretion follows a distinct circadian rhythm [2].

Chronic stress can lead to various pathologies. Burden of environmental stress can acutely or chronically alter the quantity or pattern of secretion, well described in relation to jet-lag posttraumatic stress disorder [3,4]. A bidirectional relationship exists between mood disorders and circadian rhythms. Mood disorders and depression are often associated with disrupted circadian clock-controlled responses [5,6,7]. Additionally, some chronic diseases (like hypercortisolism, adrenal insufficiency, congenital adrenal hyperplasia, Alzheimer’s disease, heart failure, etc.) are associated with changes in cortisol levels [8,9,10]. Insight into cortisol fluctuations, subdivided by sex, may contribute to patient care and therapeutic management. There is a sex-specific difference in cortisol levels and patterns [11]. Previous research has shown that males and females often report similar levels of subjective stress yet exhibit distinct physiological responses. Males typically demonstrate a more pronounced salivary cortisol response, while females show a more attenuated pattern [12]. These differences are thought to be modulated by sex hormones, with estradiol enhancing and testosterone suppressing HPA activity. Moreover, sex-related variations in cortisol secretion have been observed across the day, with females showing greater variability and, in some studies, higher morning basal levels [13]. These findings support the relevance of investigating sex as a biological variable in stress-related research.

### 1.1. Salivary Cortisol Is a Non-Invasive Stress Biomarker

Salivary cortisol is frequently used as a stress biomarker. Under physiological conditions, it exhibits a circadian pattern of secretion, with a peak in the morning—corresponding to waking—and serves an essential function of synchronizing peripheral clocks with the central molecular clock in the suprachiasmatic nucleus of the hypothalamus [14]. Salivary cortisol measurement is a common non-invasive way to determine daily fluctuations [15]. There is a high correlation between salivary cortisol levels and free cortisol levels in plasma and serum, which remains high throughout the circadian cycle and during dynamic tests [16]. Salivary cortisol and α-amylase are also associated with cardiovascular and cardiometabolic risks, with dynamic changes and altered secretion patterns serving as subclinical indicators of stress [17].

### 1.2. COVID-19 Pandemic Was Serious Stressor

The SARS-CoV-2 pandemic (COVID-19) and lockdown which included isolation and social distancing were significant stressors that brought people to novel, unpredictable and uncertain living situation [18,19]. According to the World Health Organization (WHO), the COVID-19 outbreak has resulted in the death of more than 380,000 people worldwide by the first days of June 2020. Final numbers for 2 November 2023, are 6,934,072 deaths [20]. Social isolation, loss of job and everyday life routine, an increase in conflicts inside the family due to 24-h staying together, separation from close people and even their death, and excessive consumption of alcohol are just several factors that affected people’s behavior and physiology during lockdown. All these factors, including the constant fear for the health of oneself and loved ones, strained mental health, and during COVID-19 pandemics significantly higher prevalence of depression, anxiety, insomnia, PTSD, and psychological distress were found regardless of sex, group or region [21]. The COVID-19 pandemic has affected women in unique sex-specific ways as home managers. The COVID-19 lockdown has intensified women’s domestic workload, and remote work has contributed to severe challenges [22].

### 1.3. Quarantine Measures for Foreign Medical Students in Ukraine

Due to the worldwide COVID-19 outbreak, the Ukrainian government declared a state of emergency on 12 March 2020, and a few days later, strict lockdown measures were implemented to prevent the spread of the virus. The measures included a ban on gatherings in public places and across regional borders, the closure of schools, universities and all non-essential businesses, and the instruction to work from home with very few exceptions. At that time two hundred foreign medical students of I. Horbachevsky Ternopil National Medical University, Ternopil, Ukraine (TNMU) (first to sixth year of study) was isolated within dormitory facilities and were not allowed to exit, including for walks or other outdoor activities. Self-isolation and “stay at home” measures were implemented nationwide and communicated through public officials and various communication channels.

This cross-sectional study aimed to investigate sex-specific patterns in daily salivary cortisol fluctuation and perceived stress, including physical and behavioral symptoms, among foreign students of TMNU related to the COVID-19 lockdown. Given the unique circumstances in which all participants were confined to the same dormitory in a foreign country, without access to external support systems (such as immediate family support), the sample provided a controlled setting for examining stress-related physiological and psychological responses. We hypothesized that sex would be associated with differences in both physiological (salivary cortisol) and psychological (perceived stress and related symptoms) responses to the lockdown environment. In a subsequent analysis, we explored the academic performance of these students during pre-COVID (2018/2019), COVID (2019/2020), and post-COVID academic year (2020/2021).

## 2. Materials and Methods

### 2.1. Study Design

A cross-sectional study was conducted in the second part of the May 2020, during academic year 2019/2020 and Ukrainian lockdown that had already lasted 2 months, with no indication of when it might end. The study lasted over a whole day, from waking up until bedtime, during which participants collected saliva samples and completed questionnaires (self-assessments of stress experiences and health issues). The study was approved by the Bioethics Committee of TNMU (Protocol No. 59 from 29 April 2020). The study was performed based on the principles of Declaration of Helsinki (2013). Informed consent was received from every participant.

### 2.2. Participants

The study primarily involved 200 foreign medical students enrolled in the first through sixth years at TNMU, located in Ternopil, Ukraine. Inclusion criteria for participants were as follows: compliance with lockdown requirements, signed informed consent, agreement to saliva collection, and the presence of all five requested samples. Exclusion criteria were acute illness including COVID-19 or exacerbation of chronic disease. Nationality was recorded for all participants, but it was not included in the subsequent analyses. For female participants, menstrual cycle phase was not controlled during sampling.

### 2.3. Sampling of Salivary Cortisol

Saliva was self-sampled by participants according to precise and detailed guidelines provided by the research team—in written form and via online meeting, ensuring compliance with the study protocol and standardized procedures across the sample. Sampling was performed 5 times over 1 day using Salivette (Sarstedt, Nümbrecht, Germany). Sample collection was conducted from 07:00–09:00 (t_1_) and continued in 4-h increments (11:00–13:00 (t_2_), 15:00–17:00 (t_3_), 19:00–21:00 (t_4_), 23:00–01:00 (t_5_)). All participants were instructed to adhere to dietary and activity restrictions prior to sampling. They were carefully advised to gently chew on the provided cotton roll to stimulate saliva flow rate and to follow the time frames for saliva collection; those who did not provide a sample during the morning collection window, when participants were expected to be awake due to scheduled academic obligations, were not taken into consideration for the research. All students who provided samples were recruited in this manner, irrespective of their habitual circadian rhythms. Samples were analyzed for cortisol content (ng/mL) using the Cortisol ELISA Kit (Abcam, Cambridge, UK; ab154996) and microplate photometer (Multiscan FC-357, Thermo Fisher Scientific, Waltham, MA, USA).

#### Mathematical Saliva Cortisol-Based Clustering of Participants

Given the interindividual variability in salivary cortisol levels and the absence of clearly identifiable subgroups based on initial screening, the obtained cortisol concentrations were subjected to Mathematical Modelling using Mathematica software (ver. 12.0, Wolfram Research, Inc., Champaign, IL, USA). Clustering was employed as an exploratory method to identify natural groupings in the data, allowing detection of meaningful patterns across multiple variables without imposing predefined categories. It should be noted that the menstrual cycle phase was not included as a covariate in this data preparation or cluster assignment. This factor was treated as a source of non-controlled biological variability. Data set review revealed that several samples (6 male and 2 female) were incomplete (e.g., male sample no. 51 missed 2nd saliva cortisol measurement). To minimize data loss while maintaining dataset integrity, missing measurements were reconstructed from existing data using a median of similar data (see [23,24]). After outlier elimination using the MinPts and EPS parameters of the Density-Based Spatial Clustering of Applications with Noise (DBSCAN) method [25,26,27,28,29], a set of 90 males and 87 females remained. The data was normalized by the simple linear transformation into the unit hypercube [0, 1]^5^ [29]. In order to perform the clustering procedure, the most appropriate number of clusters (MAPart) was determined using several known indices: Calinski–Harabasz (CH) index, Davies–Bouldin (DB) index, Simplified Silhouette with Criterion (SSWC), and Dunn index [30]. The partition with the largest CH-index, the smallest DB-index, the largest SSWC-index and the smallest Dunn-index was deemed MAPart. While for the male dataset, the DB index was unusable, other indices suggested 5-partition as the MAPart. The choice of MAPart for the female data set was inconsistent; therefore, a 5-partition was chosen again. Finally, the clustering procedure was carried out. It was performed was performed separately for male and female participants. Samples from the data sets were grouped into spherical clusters applying the Least Square distance-like function
$d_{LS}\left(x,y\right)={\left\|x-y\right\|}^{2}$. Cluster centroid and its saliva cortisol concentrations for one day, variance (σ^2^), and standard deviation (SD) were determined. Beyond index-based selection, cluster validity was assessed through biological interpretability and visual inspection. Additionally, area under the curve (AUC), amplitude, and fixed-time slope of the cortisol curve were calculated. AUC was computed with respect to ground using the trapezoidal rule, amplitude was calculated as
$\frac{C_{\max}\left(cortisol\right)-C_{\min}\left(cortisol\right)}{2}$, and the fixed-time slope was analyzed between measurement points 2 and 4 to observe a return to baseline values. It was derived using the formula
$\frac{{C(cortisol)}_{t_{4}}-{C(cortisol)}_{t_{2}}}{t_{4}-t_{2}}$, where the time difference between ***t*_4_** and ***t*_2_** was 8 h.

### 2.4. Instruments for Assessing Perceived Stress and Health Concerns

The second part of the study included the self-reported questionnaire “Perceived Stress Scale-10” (PSS-10) [31] and an unvalidated complement questionnaire based on the The Kaiser Family Foundation (KFF) Health Tracking Poll—July 2020 [32] to assess the impact of chronic stress on lifestyle and health issues during the previous two months.

The PSS-10 is a widely used psychological instrument for measuring perceived stress. Developed by Sheldon Cohen and colleagues in 1983 [31], it evaluates the extent to which individuals perceive their lives as being unpredictable, beyond their control, and excessively demanding [33,34,35,36]. It is validated across diverse populations and cultures and correlates with health outcomes, such as cortisol levels, sleep quality, and mental health [37]. The PSS-10 contains 10 items rated on a 5-point Likert scale (0 = never to 4 = very often). Items reflect feelings and thoughts during the last month. The PSS-10 total score ranges from 0 to 40, with higher values reflecting greater perceived stress. Based on the total score, stress levels can be categorized as follows: 0–13 (Low), 14–26 (Moderate), and 27–40 (High). The data completeness for the PSS-10 was 99.6%, with only 7 individual items missing across the entire dataset of 177 participants. Specifically, missing responses occurred on item 5 (*n* = 2), item 6 (*n* = 1), item 7 (*n* = 1), and item 8 (*n* = 3). To maintain the full sample size (*N* = 177), subscale-based pro-rating was applied, where missing values were replaced by the mean of the available items within the respective subscale (Perceived Helplessness for item 6; Self-Efficacy for items 5, 7, and 8).

The analysis also covered the two subscales of the PSS-10: Perceived helplessness (PH) (items 1, 2, 3, 6, 9, 10)—measuring an individual’s feelings of a lack of control over their circumstances or their own emotions or reactions; and Lack of self-efficacy (LSE) (items 4, 5, 7, 8)—measuring an individual’s perceived inability to handle problems.

The questionnaire based on the KFF Health Tracking Poll was used as a complement exploratory tool to the PSS-10. It is a non-validated, expert-designed questionnaire developed for the U.S. public opinion research context [32]. It consists of 10 binary (yes/no) items designed to capture self-reported physical and behavioral symptoms potentially related to stress, such as sleep disturbances, appetite changes, headaches, difficulties with emotional regulation, and increased substance use. Although not standardized for clinical use, it provides several useful insights into population-level trends in perceived health and well-being, and offers a broader perspective on how stress may manifest through somatic and behavioral symptoms in everyday life [38]. Its inclusion enables the identification of symptom patterns that might otherwise go undetected, especially in non-clinical populations.

#### PSS-10 Scale Validation and Factor Extraction

The construct validity of the PSS-10 scale was assessed using Exploratory Factor Analysis (EFA). All analyses were performed using Jamovi statistical software (Version 2.6; Sydney, Australia), utilizing the ‘psych’ package for R [39,40,41]. Demographic variables (e.g., gender, cluster group) were recorded but not included in the factor analysis.

To ensure the data’s suitability for EFA, two primary assumption checks were performed: (a) the Kaiser–Meyer–Olkin (KMO) measure equal or greater than 0.5 was used to test sampling adequacy; (b) the Bartlett’s Test of Sphericity with *p* < 0.001 was conducted to confirm the presence of sufficient correlations within the item matrix. The EFA was performed using Principal Axis Factoring (PAF) with varimax rotation and the criterion of eigenvalue greater than 1.00.

### 2.5. Students’ Academic Success

Academic performance data (grade point averages) were obtained directly from the official records of the TNMU, ensuring that the results reflect verified institutional outcomes. The grades were obtained for each participant for three academic years: the year preceding the study—2018/2019 (pre-COVID), the year during which the study was conducted—2019/2020 (COVID), and the subsequent academic year—2020/2021 (post-COVID). It should be noted that academic performance data for the 2018/2019 academic year were not available for all participants, as some students were in their first year of study during that period. This was only exploratory analysis, and we find that any correlation (i.e., Spearman’s) to cortisol daily fluctuation pattern is not justified due to cross-sectional nature of the study, and we got insight into cortisol daily fluctuation during only one day.

### 2.6. Statistical Analysis

Statistical analysis of survey results was performed using Prism software, version 10.6.1 (GraphPad Software LLC, Boston, MA, USA). Quantitative parameters were expressed according to their distributions, assessed using the Shapiro–Wilk test—variables with normal distribution were presented as mean ± standard deviation (Mean ± SD), whereas those with non-normal distribution were presented as median (lower quartile; upper quartile) [Me (Lq; Uq)]. Qualitative parameters were summarized as frequencies and percentages. Regarding evaluation of cluster validity, construct validity was first established on the entire aggregated sample using EFA, providing a robust common structure. As the study’s primary objective is descriptive, cluster-based sensitivity was addressed by comparing the mean scores derived from this validated factor structure across our defined clusters using ANOVA, rather than pursuing complex measurement invariance modeling. Comparisons among three or more normally distributed groups were performed using ANOVA with Tukey’s post hoc test. For non-normally distributed variables across three or more groups, the Kruskal–Wallis test was applied. Two independent groups with normal distribution were compared using the t-test, whereas those with non-normal distribution were compared using the Mann–Whitney U test. Two dependent groups with non-normal distribution were compared using Wilcoxon matched-pairs signed rank test. Associations between categorical variables were assessed using χ^2^ test for comparisons between sexes and Fisher’s exact test in comparisons between clusters due to small sample size of clusters. Correlations were assessed using Spearman’s nonparametric correlation. Statistical significance was set at *p* < 0.05. Effect sizes were reported for all significant comparisons: Cohen’s D for *t*-tests, Vargha-Delaney *A* for Mann-Whitney U tests, Cramer’s V for Fisher’s exact tests, and rank-biserial correlation (r) for Wilcoxon matched-pairs tests. Post-hoc power analysis revealed that the achieved power (1 − β) for primary sex-based comparisons ranged from 0.65 to 0.75, while for the cluster-based Kruskal-Wallis analysis, it exceeded 0.99. For within-group academic success comparisons with smaller effect sizes (r approximately 0.2), the achieved power was in the range of 0.3 to 0.4.

## 3. Results

Of the initial 200 students invited to participate, 177 provided valid survey responses and salivary samples, which were included in the final analysis—90 males and 87 females, all foreign students who, at the time of the study, had been confined to the dormitory for two months, with no clear indication of when the lockdown would be lifted, and under conditions of considerable uncertainty regarding the near future (Figure 1). Median age of all participants was 20 (19; 22). At the time of sample collection, none of the participants exhibited symptoms of respiratory infection. For female participants, menstrual cycle phase and the use of hormonal contraceptives were not controlled or recorded at the time of salivary sampling. This was considered a potential confounding factor and is further addressed in the Section 4.5.

### 3.1. Salivary Cortisol

Salivary cortisol was sampled five times throughout the day, starting at awakening (at 07:00–09:00) and ending at bedtime (at 23:00–01:00). The characteristics of the cortisol curve for all participants, including its AUC, amplitude, slope, and diurnal variation, were analyzed. Furthermore, sex-specific analyses were performed to examine potential differences in cortisol curve characteristics between male and female participants. There were no statistically significant sex-specific differences in AUC, amplitude or slope of the salivary cortisol curve (Figure 2). When observing the cortisol curve of all participants together, the highest salivary cortisol concentration was recorded at the second measurement point (at 11:00–13:00), which represents a shift from the typical cortisol pattern, where the peak concentration occurs in the early morning before awakening (corresponding to the first measurement point or even earlier) and gradually decreases throughout the day. When examining cortisol levels by sex, higher concentrations were observed in males. Statistically significant differences between males and females were found at the third and fourth measurement points, where males had higher cortisol levels (t_3_: *p* = 0.021, U = 3130, t_4_: *p* = 0.009, U = 3025) (Figure 2).

#### Mathematical Clustering of Participants According to the Salivary Cortisol Daily Fluctuation

Using mathematical clustering to detect meaningful patterns, five distinct subgroups were identified among male and female participants based on similarities in their daily salivary cortisol fluctuation profiles (Figure 3), although it should be highlighted that the menstrual cycle phase was not included as a variable in the data modeling for female participants. Notably, female cluster 4 (π_4_) included only two participants, which is an insufficient sample size for statistical analysis. This cluster may be considered as an outlier. The characteristics of the cortisol curves for all clusters were also analyzed (AUC, amplitude, and slope) (Figure A1).

Cluster 1 (π_1_) grouped 51.1% males and 66.7% females. This cluster exhibited minimal variation in cortisol levels throughout the day. AUC, amplitude of cortisol fluctuation, and slope were similar for males and females. Cluster 2 (π_2_) included 14.4% male and 6.9% female participants. AUC of cortisol curve was similar for both sexes. Their cortisol curve showed a typical diurnal pattern, with peak levels in the morning, a gradual decline throughout the day, and a nadir at night. The amplitude of cortisol fluctuation was significantly lower in males compared to females (Mann-Whitey U test, *p* = 0.029, U = 14), suggesting a less pronounced decline in cortisol concentration throughout the day in male participants which was visible in the values of the slopes. The third cluster (π_3_) contained 18.9% of males and 16.1% of females. Their cortisol curves showed a shift in peak cortisol levels, with the maximum occurring at the second measurement point rather than the first. This represents an atypical diurnal cortisol pattern. AUC was similar between males and females. In comparison to the females’ curve, the slope of the males’ curve was significantly lower (Mann-Whitney U test, *p* = 0.019, U = 60). The amplitude of fluctuation was also significantly lower in males (Mann-Whitney U test, *p* = 0.01, U = 55). The male and female clusters 4 (π_4_) included 11.1% of males and only 2 females (2.3%). Their curves similarly showed a maximal peak at the fourth measurement point (corresponding to 19:00–21:00). These π_4_-clusters represented an irregular (atypical) cortisol pattern. The sample size of female π_4_ did not allow statistical analyses. Clusters 5 (π_5_) also exhibited an irregular (atypical) cortisol pattern. These clusters included 4.4% male and 8% female participants. Males had higher AUC than females (Kruskal-Wallis and Dunn’s post hoc test, *p* < 0.001) (Figure A1).

### 3.2. Perceived Stress and Health Concerns

The construct validity of the PSS-10 scale was assessed using Exploratory Factor Analysis (EFA). KMO and Bartlett’s test of Sphericity confirmed adequacy of the data for EFA. KMO was 0.787, and Bartlett’s Test of Sphericity was significant (χ^2^ = 496, df = 45, *p* < 0.001). EFA was conducted using PAF with Varimax rotation. Two factors were identified (factor 1 including items from subscale PH and factor 2 including items from subscale LSE) and their loadings are shown in Table 1. Those two factors together explained 44.3% of the total variance.

The average total PSS-10 score was significantly higher for women than men (Table 2). 46.7% of males and 63.2% of females had total PSS-10 scores which correspond to moderate levels of perceived stress. Low perceived stress was reported by 51.1% of men and 32.2% of women, whereas high perceived stress was reported by 2.2% of men and 4.6% of women. Total PH-score of females was significantly higher than the one in males, while LSE did not differ between sexes (Table 2). The overall score of items 2 and 9, which was significantly higher in females, contributed to the sex-specificity in the PH subscale (Table 3).

After salivary cortisol-based clustering, mentioned stratification in magnitude of perceived stress was altered—these scores indicated that respondents from male clusters with predicted daily cortisol pattern (π_1_ and π_2_) and from male π_5_-cluster were perceiving low stress levels and all female clusters were perceiving moderate levels of stress (Table 2). Sex-specificity in PH score was lost after clustering.

When the distribution of responses to the PSS-10 items was compared between males altogether and females altogether, no sex-specific difference was found (Figure A2). After salivary cortisol-based clustering, few sex-specific differences in the distribution of responses to PSS-10 were observed, localized within the PH subscale (items 1, 2, and 9) where females had significantly differently distributed responses resulting in higher scores (Table 3). Cluster-based approach elucidated a minor sex-specific difference within the LSE subscale; specifically, males from π_2_ scored higher on item 5 (tending toward “often”) than females from the same cluster (tending toward “sometimes”) (Table 3).

In the context of stress experience research, another sex-specific difference emerged among the clusters: cortisol AUC significantly correlated with PH scores within female clusters exhibiting aberrant daily cortisol patterns. Specifically, this correlation was moderate positive for the π_3_ cluster (Spearman’s r = 0.56, *p* = 0.039, r^2^ = 0.31), whereas it was strong negative for the π_5_ cluster (Spearman’s r = −0.79, *p* = 0.048, r^2^ = 0.62). Conversely, no such associations were observed among male clusters. Due to its small sample size, the female π_4_ cluster was excluded from this analysis.

The KFF Health Tracking Poll served as a complement exploratory tool to the PSS-10 to assess perceived health and well-being of the participants, and it yielded several noteworthy insights. There was no statistically significant difference in the frequency of answers between sexes or clusters; therefore, answers were analyzed together for all participants (Table 4). The table shows the frequency of symptoms related to stress, emotional, and physical difficulties. Most participants do not report symptoms, but there is a large proportion who still report challenges (around one third of all participants).

### 3.3. Academic Success 2018/2019–2020/2021

The overall academic success of respondents was obtained from the official records of the TNMU. It was analyzed in the academic year preceding this study (2018/2019, pre-COVID), in the academic year during which the study was conducted (2019/2020, COVID), and in the year after (2020/2021, post-COVID). The data is shown in Table A1. Data for the pre-COVID academic year was not available for respondents who were in their first year of study at that time. The goal of this analysis was to assess potential changes or trends in academic performance over time that could have been driven by lockdown-induced stress.

Our findings reveal a notable upward trend in grades during and after the pandemic period. Females showed a significant improvement in grades in COVID year compared to pre-COVID (*p* = 0.022), with further gains observed in the post-COVID year (vs. COVID year: *p* = 0.025, vs. pre-COVID year: *p* < 0.001). Males had significantly better grades in post-COVID year compared to pre-COVID year (*p* < 0.001).

After cortisol-based clustering of the participants, significant changes were observed only within π_1_-clusters: in comparison pre-COVID vs. COVID year, only females showed significantly better grades (*p* = 0.009), suggesting sex-specificity in academic success; in comparison COVID vs. post-COVID year, both males and females within the π_1_-cluster showed significant improvement in grades (males: *p* = 0.033, females: 0.004); and comparison of pre-COVID vs. post-COVID year revealed that both males and females within the π1-cluster had significantly better grades (males: *p* = 0.008, females: *p* < 0.001).

## 4. Discussion

This study set out to evaluate the sex-specific adaptation to the unprecedented stressors of the COVID-19 pandemic and subsequent lockdown among foreign medical students at TNMU. We hypothesized that male and female students would exhibit distinct physiological (salivary cortisol) and psychological (PSS-10 and KFF health tracking poll) response patterns to the isolation and academic shifts of the lockdown period. Furthermore, we aimed to examine the longitudinal academic performance of these students across three distinct academic years (pre-COVID, COVID, and post-COVID) to contextualize the potential impact of the pandemic environment on their educational outcomes.

### 4.1. Sex-Specific Patterns and Mathematical Clustering of Salivary Cortisol

Our comparative analysis revealed significant sex-specific differences in physiological stress markers. Specifically, male participants exhibited significantly higher cortisol levels compared to females, with statistical significance reached at the third (15:00–17:00 h) and fourth (19:00–21:00 h) sampling points. This finding is in accordance with previous reports [12]. While initial observations of the aggregate data showed an unexpected cortisol peak between 11:00 and 13:00 h for all participants, mathematical clustering was employed to further investigate this phenomenon. This modeling revealed five distinct secretory patterns, suggesting that while two-thirds of the cohort (around 70% of participants from clusters π_1_ and π_2_) followed a conventional diurnal rhythm [2], the remaining one-third displayed atypical patterns that distorted the group’s mean curve. These findings highlight that robust underlying variability exists within both sexes—a phenomenon well-documented in the literature regarding sex-specific HPA axis responses [11,42,43], which may be further influenced by methodological factors such as the lack of control for individual ‘morningness’ [44] or the menstrual cycle phase in female participants [45]. Furthermore, given the cross-sectional nature of this study, it is not possible to determine pre-lockdown cortisol dynamics or the specific longitudinal stressors to which these individuals were exposed. While our clustering revealed that approximately one-third of the cohort exhibited atypical cortisol fluctuations, it remains unclear how these proportions compare to pre-pandemic conditions or whether these patterns represent a transient or a chronic adaptation to the lockdown environment.

### 4.2. Perceived Stress and Somatic Symptoms

Analysis of the PSS-10 subscales provided further insight into the psychological architecture of the students’ stress. The total PSS-10 score falls within the expected range based on age-related norms reported in the literature [46]. Both sexes operated within the moderate stress range, which may reflect the preserved structure and cohesion of the student community during lockdown. The availability of virtual communication and continuity of daily routines likely mitigated the psychological impact, preventing the disintegration of social structure typically observed in more chaotic scenarios [47]. However, male participants reported significantly lower total PSS-10 scores and lower Perceived Helplessness (PH) compared to females. This sex-specific pattern may reflect differences in stress appraisal or coping mechanisms. Epidemiological studies consistently show that women are more frequently diagnosed with anxiety and depression than men, with female-to-male prevalence ratios often reaching 2:1. Sex differences in coping strategies may partly explain this disparity, as women tend to rely more on emotion-focused coping—such as rumination and self-blame—which is associated with higher levels of psychological distress. In contrast, men are more likely to engage in problem-focused coping [48,49,50]. The magnitude of these differences was supported by small-to-medium effect sizes, suggesting that the sex-specific divergence in emotional vulnerability is a robust feature of this population’s response to lockdown. These findings indicate that despite their higher physiological cortisol output, male students maintained a significantly stronger sense of control. In contrast, the absence of significant differences in Lack of Self-Efficacy (LSE) across sexes and clusters suggests that the pandemic exerted a uniform pressure on the students’ confidence in their personal coping abilities, regardless of their biological profile or perceived helplessness.

To provide a broader context for the students’ wellbeing, an exploratory KFF Health Tracking poll was used to screen for common somatic and behavioral symptoms across the entire sample. The descriptive results for all participants revealed that approximately one-third of them experienced challenges. These observations suggest that the moderate psychological stress captured by the PSS-10 was accompanied by tangible physical and emotional toll for a significant portion of the student population, reinforcing the need for holistic support during prolonged periods of isolation.

### 4.3. Sex-Specific Correlation Between Physiological and Psychological Stress Markers

A further exploratory analysis within the female clusters suggested divergent associations between cortisol output and psychological distress. Specifically, in the π_3_ cluster, cortisol output was found to correlate positively with PH, whereas the π_5_ cluster showed a strong inverse relationship. While these results must be interpreted with extreme caution due to the limited number of participants in these subgroups, they offer an interesting insight into possible individual differences in stress adaptation. It is well-recognized that these differences arise from the complex interplay genetic predispositions and individual life histories, which ‘program’ the HPA axis and shape how an individual perceives and biologically integrates environmental challenges [51]. Furthermore, since the phase of the menstrual cycle was not controlled, it remains a source of non-modeled biological variability that could influence individual cortisol output and its correlation with perceived stress scores. Nevertheless, this divergence might point toward a ‘decoupling’—a separation of physiological and psychological stress responses—within the π_5_ subgroup. These specific associations were notably absent in all male clusters, suggesting the possibility that this biological-psychological interaction might be more pronounced in females under the specific conditions of the lockdown.

### 4.4. Academic Success Trends Under Lockdown-Induced Stress

The longitudinal look at academic performances (which were verified institutional outcomes, should be considered exploratory, as academic success was not directly correlated with other psychological or physiological variables in this study. The observed upward trend in grades—most evident in the larger π_1_ clusters—suggests a generalized resilience, though it may also reflect altered assessment criteria during the transition to online education, as suggested by Gonzales et al. [52]. Furthermore, while the cortisol-based clustering provided a unique framework for analysis, it is important to acknowledge that our one-day sampling protocol may not fully capture the complete diurnal cortisol patterns throughout the prolonged lockdown period. Despite these limitations, the fact that students maintained or improved their academic standing amidst reported somatic symptoms and reported stress scores points toward a notable capacity for adaptation. This highlights the complex relationship between biological stress profiles and functional outcomes in high-pressure environments like medical school [53].

### 4.5. Limitations

This study has limitations that should be acknowledged. First, its cross-sectional design limits the ability to draw causal inferences about the relationship between stress-related variables and academic performance. Longitudinal data on stress biomarkers and psychological scores would be necessary to determine the directionality and stability of these relationships over time. Second, the lack of pre-lockdown cortisol measurements prevents direct comparison with baseline physiological stress levels, making it difficult to assess the specific impact of the lockdown period. Third, although the sample size was sufficient for the analyses conducted, the sample was particular, comprising foreign students at a single university during a unique global event, which limits the generalizability of the findings to broader student populations. Future research should aim to include more diverse and representative samples, as well as longitudinal designs that capture stress dynamics before, during, and after major stressors. And, lastly, menstrual cycle phase was not controlled among female participants. The luteal phase is known to influence cortisol secretion [43], and its variability could have contributed to interindividual differences observed in the data. However, the aim of this study was not to examine phase-specific effects but rather to identify overall patterns of cortisol variation. Future research should consider controlling or recording cycle phase to reduce potential bias and improve interpretability.

## 5. Conclusions

This study explores the complex interplay between psychological perception, biological stress profiles, and academic outcomes among foreign medical students during the COVID-19 lockdown. Mathematical modeling has proven to be a useful tool for identifying subgroups within a large population of participants. While female students reported higher levels of perceived stress, the use of cortisol-based clustering revealed diverse physiological phenotypes. The exploratory findings in female subgroups suggested a potential decoupling between psychological distress and cortisol output. However, these biological patterns remain preliminary due to limited subgroup sizes and the lack of control for menstrual cycle phases. Despite these challenges, a notable upward trend in academic performance was observed, particularly within the stable π_1_ clusters. While this success may partially reflect adapted online assessment criteria, it underscores a generalized academic resilience. Ultimately, these results highlight the students’ capacity to maintain functional outcomes under environmental pressure and emphasize the need for institutional support that recognizes the diverse stress phenotypes within the student population.

## Figures and Tables

**Figure 1 life-16-00054-f001:**
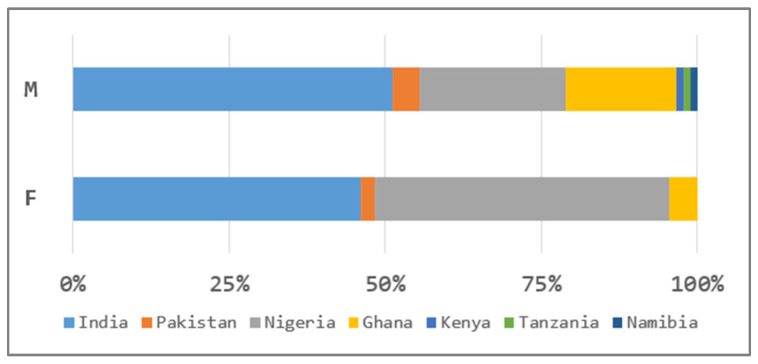
Percentage distribution of participants by nationality; M—males (*n* = 90), F—females (*n* = 87).

**Figure 2 life-16-00054-f002:**
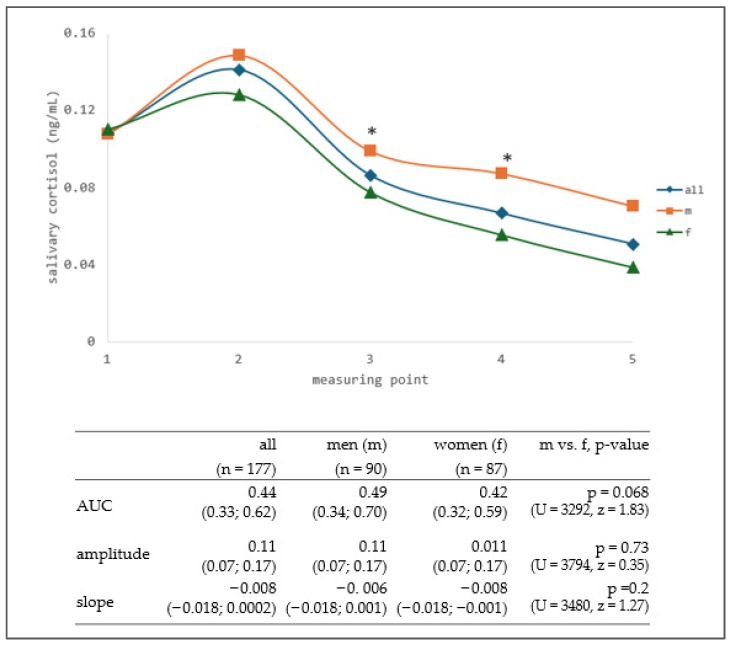
Daily fluctuation of salivary cortisol in all participants, and in men and women (median value), with characteristics of salivary cortisol curve. Salivary cortisol was measured at five time points throughout the day: (1) 07:00–09:00, (2) 11:00 –13:00, (3) 15:00–17:00, (4) 19:00–21:00, and (5) 23:00–01:00. AUC (area under the curve), amplitude, and slope are presented as median with interquartile range (Lq; Uq). *—*p* < 0.05, male vs. female (Mann–Whitney U test).

**Figure 3 life-16-00054-f003:**
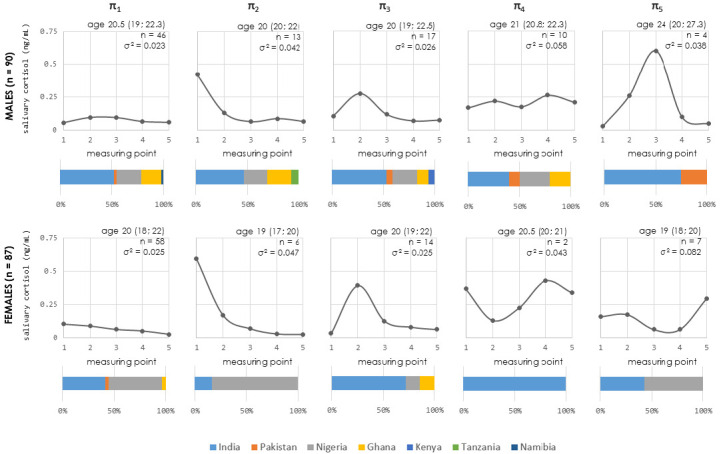
Comprehensive profiles of clusters (π_1_–π_5_) based on salivary cortisol daily fluctuations. Main panels display the centroid cortisol daily fluctuations based on mathematical modeling. Internal labels indicate cluster sample size (*n*), median age (with interquartile range), and variance of centroid concentrations (σ^2^). Sub-panels below each main graph illustrate the participant distribution by nationality (e.g., the predominant representation of India and Ghana in cluster π_1_). Salivary cortisol was measured at five standardized intervals (measurement points throughout one day): (1) 07:00–09:00, (2) 11:00 –13:00, (3) 15:00–17:00, (4) 19:00–21:00, and (5) 23:00–01:00. Note: the menstrual cycle phase was not included as a variable in the data modeling for female participants.

**Table 1 life-16-00054-t001:** Factor loadings by exploratory factor analysis for Perceived stress scale-10.

Question		Factor Loading
In the last month, how often have you…		
	Factor 1. Perceived helplessness	
1	been upset because of something that happened unexpectedly?	0.755
2	felt that you were unable to control the important things in your life?	0.699
3	felt nervous and “stressed”?	0.745
6	found that you could not cope with all the things that you had to do?	0.619
9	been angered because of things that were outside of your control?	0.644
10	felt difficulties were accumulating so high that you could not overcome them?	0.593
	Factor 2. Lack of self-efficacy	
4	felt confident about your ability to handle your personal problems?	0.681
5	felt that things were going your way?	0.715
7	been able to control irritations in your life?	0.500
8	felt that you were on top of things?	0.632

**Table 2 life-16-00054-t002:** Total scores of Perceived stress scale-10 (PSS-10), and Perceived helplessness (PH) and Lack of self-efficacy (LSE) subscales of male and female participants overall or grouped in clusters (π_1–5_) according to the daily fluctuation of their salivary cortisol. All data collected in the second part of May 2020, two months after the start of the lockdown.

	**Males**	**Male Clusters**				
		**π_1_**	**π_2_**	**π_3_**	**π_4_**	**π_5_**
Total scores	(*n* = 90)	(*n* = 46)	(*n* = 13)	(*n* = 17)	(*n* = 10)	(*n* = 4)
PSS-10 (mean ± SD)	* 14 ± 6	13.3 ± 5.5	13.5 ± 5.5	16.2 ± 6.8	14.7 ± 8.1	10.8 ± 7.4
PH (median [Lq; Uq])	^†^ 6.5 (4; 10)	6 (3; 10)	8 (5; 10)	7 (5; 17)	9 (3; 14)	7 (2; 9)
LSE (median [Lq; Uq])	6 (4; 9)	6 (4; 9)	5 (3; 8)	6 (4; 9)	6 (3; 7)	5 (1; 8)
	**Females**	**Female Clusters**				
		**π_1_**	**π_2_**	**π_3_**	**π_4_**	**π_5_**
Total scores	(*n* = 87)	(*n* = 58)	(*n* = 6)	(*n* = 14)	(*n* = 2)	(*n* = 7)
PSS-10 (mean ± SD)	* 16 ± 6	15.8 ± 6.1	16.5 ± 4.8	15.1 ± 4.4	15 ± 1.4	19.3 ± 9.9
PH (median [Lq; Uq])	^†^ 9 (6; 13)	9 (6; 13)	10 (8; 13)	8 (6; 9)	8 (8; 8)	9 (7; 21)
LSE (median [Lq; Uq])	6 (5; 9)	6 (4; 9)	7 (2; 9)	7 (6; 10)	7 (6; 8)	6 (5; 11)

* *t* test (males vs. females): *p* = 0.022 (Cohen’s D = 0.35), ^†^ Mann-Whitney U test (males vs. females): *p* = 0.012 (U = 3063, z = 2.51; Vargha-Delaney *A* effect size = 0.39); *n*—number of responses; stress categories according to the total PSS-10 score: 0–13 (Low), 14–26 (Moderate), and 27–40 (High). Missing items on the PSS-10 scale for 7 participants were reconstructed using pro-rating to maintain the total sample size.

**Table 3 life-16-00054-t003:** Average scores (median with interquartile range) of the responses on the Likert scale (where 0 = never, 1 = almost never, 2 = sometimes, 3 = fairly often, 4 = very often) to 10 items of questionnaire Perceived Stress Scale-10 presented for men and women overall, and grouped in clusters (π_1–5_) according to the daily fluctuation of their salivary cortisol. All data collected in the second part of May 2020, two months after the start of the lockdown. Items are also grouped according to the perceived helplessness (PH) and lack of self-efficacy (LSE) subscales.

	**Males**	**Male**	**Clusters**			
**Question**		**π_1_**	**π_2_**	**π_3_**	**π_4_**	**π_5_**
In the last month how often have you…	(*n* = 90)	(*n* = 46)	(*n* = 13)	(*n* = 17)	(*n* = 10)	(*n* = 4)
PH subscale						
(1) been upset because of something that happened unexpectedly?	1 (0; 2)	^a^ 1 (0; 1.3)	1 (0; 2)	2 (0; 2)	1.5 (0; 3.3)	1.5 (0.25; 2)
(2) felt that you were unable to control the important things in your life?	^f^ 1 (0; 2)	^b^ 0 (0; 2)	1 (0; 2)	^c^ 1 (0; 2.5)	0.5 (0; 2.3)	1 (0.25; 1.8)
(3) felt nervous and “stressed”?	1 (0; 2)	1 (0; 2)	2 (1; 2)	2 (1; 3)	1.5 (0; 2)	0.5 (0; 1)
(6) found that you could not cope with all the things that you had to do?	1 (0; 2)	1 (0; 2) [*n* = 44]	1 (0.5; 2)	2 (0.5; 3)	2 (0.75; 2.3)	1 (0.25; 1.8)
(9) been angered because of things that were outside of your control?	^g^ 2 (0; 2)	1 (0; 2)	2 (0; 2)	2 (0.5; 3)	2 (0.75; 2.5)	^d^ 1 (1; 1)
(10) felt difficulties were accumulating so high that you could not overcome them?	1 (0; 2)	1 (0; 2)	1 (1; 2)	1 (0; 2.5)	1 (0.75; 2.3)	1 (0.25; 2.5)
LSE subscale						
(4) felt confident about your ability to handle your personal problems?	3 (2; 4)	3 (2; 4)	3 (3; 4)	3 (2; 4)	3 (3; 4)	3.5 (3; 4)
(5) felt that things were going your way?	3 (2; 3)	2 (2; 3) [*n* = 44]	^e^ 3 (2; 3)	2 (1; 3)	3 (2; 3.3)	3.5 (2.3; 4)
(7) been able to control irritations in your life?	3 (1; 3)	2 (1; 3)	3 (2; 3.5)	3 (2; 3)	3 (1; 3)	2 (0.25; 3.8)
(8) felt that you were on top of things?	2 (2; 3)	2 (2; 3) [*n* = 45]	2.5 (2; 3) [*n* = 12]	2 (1; 3)	2.5 (1.8; 3.3)	2.5 (1.3; 3.8)
	**Females**	**Female**	**Clusters**			
**Question**		**π_1_**	**π_2_**	**π_3_**	**π_4_**	**π_5_**
In the last month how often have you…	(*n* = 87)	(*n* = 58)	(*n* = 6)	(*n* = 14)	(*n* = 2)	(*n* = 7)
PH subscale						
(1) been upset because of something that happened unexpectedly?	2 (0; 2)	^a^ 2 (0; 2.3)	2 (1.8; 2)	1 (0; 2)	1 (0; 2)	2 (0; 4)
(2) felt that you were unable to control the important things in your life?	^f^ 2 (0; 2)	^b^ 1.5 (0; 2)	1.5 (1; 2.3)	^c^ 1.5 (0; 2)	2 (2; 2)	1 (0; 2)
(3) felt nervous and “stressed”?	2 (1; 2)	2 (1; 2)	2.5 (1.8; 3)	1 (0; 2)	1.5 (1; 2)	2 (0; 4)
(6) found that you could not cope with all the things that you had to do?	2 (0; 2)	1 (0; 2)	2 (0.75; 3)	2 (0; 2)	1.5 (1; 2)	2 (0; 3)
(9) been angered because of things that were outside of your control?	^g^ 2 (1; 2)	3 (2; 3) [*n* = 57]	2 (1; 2.3)	2 (1; 2)	1.5 (1; 2)	^d^ 2 (2; 3)
(10) felt difficulties were accumulating so high that you could not overcome them?	1 (0; 2)	1 (1; 2)	1 (0; 1.5)	1 (0; 2)	0.5 (0; 1)	3 (0; 3)
LSE subscale						
(4) felt confident about your ability to handle your personal problems?	3 (2; 4)	3 (2; 4)	2.5 (1; 4)	2.5 (1.8; 3)	2 (1; 3)	3 (1; 4)
(5) felt that things were going your way?	2 (1; 3)	2 (2; 3)	^e^ 2 (1; 3)	2 (1.8; 3)	2.5 (2; 3)	2 (1; 2)
(7) been able to control irritations in your life?	2 (2; 3)	3 (2; 3)	2.5 (2; 4)	2 (1; 3)	1 (0; 2)	2 (1; 3)
(8) felt that you were on top of things?	2 (1; 3)	2 (1; 3) [*n* = 57]	3 (1.8; 3.3)	2 (1; 3)	3.5 (3; 4)	2 (1; 2)

*n* = number of responses; [*n*] indicates specific number of responses due to missing value. Fisher’s exacts test (due to small sample size) on response frequency distribution (male cluster vs. female cluster): ^a^ *p* = 0.012 (Cramer’s V = 0.34), ^b^ *p* = 0.049 (Cramer’s V = 0.3)), ^c^ *p* = 0.025 (Cramer’s V = 0.58), ^d^ *p* = 0.006 (Crmer’s V = 1), ^e^ *p* = 0.037 (Cramer’s V = 0.74); χ^2^ test on response frequency distribution (males vs. females) *p* > 0.05; t test on particular item score (males vs. females): ^f^ *p* = 0.019 (Vargha-Delaney *A* effect size = 0.4), ^g^ *p* = 0.03 (Vargha-Delaney A effect size = 0.41).

**Table 4 life-16-00054-t004:** Answer frequency (%) to the items from KFF Tracking Health Poll for all participants combined, regardless of sex (N = 177) ^§^. All data collected in the second part of May 2020, two months after the start of the lockdown.

	Question	No	Yes	NA
1	Do you experience sleep problems?	80.8	19.2	0.0
2	Do you experience poor appetite?	93.8	5.1	1.1
3	Do you experience overeating?	88.1	11.9	0.0
4	Do you have more frequent headaches?	85.3	14.7	0.0
5	Do you have more frequent stomachaches?	95.5	4.5	0.0
6	Do you experience tamper control (angrier) difficulties?	85.9	13.6	0.6
7	Do you experience increased alcohol use?	87.6	0.6	11.9
8	Do you experience increased drug use?	92.7	0.0	7.3
9	If you have some illness, do you experience worsening of your conditions?	93.8	4.0	2.3
10	Any of the above listed	62.1	37.9	0.0

NA—non applicable; ^§^ as no statistically significant differences in response frequencies were observed between men and women, and the distributions were comparable in magnitude.

## Data Availability

The raw data supporting the findings of this study will be available from the corresponding author upon reasonable and justified request, in order to maintain scientific integrity and proper contextual understanding.

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
