# Peer review of "Sex-Specific Patterns of Cortisol Fluctuation, Stress, and Academic Success in Quarantined Foreign Medical Students During the COVID-19 Lockdown"

_life, 2025, doi:10.3390/life16010054_

Round 1

Reviewer 1 Report

Comments and Suggestions for Authors

Thank you for the chance of assessing this manuscript. I am in the opinion of studies related to COVID-19 still have a lot to offer, mainly because scholar can build upon these lessons and applied methodologies for alike situations. My assessment follows the structure of the paper, with a brief overview of it.

  • General assessment: As I have said, the paper has many strengths, and my overall assessment is positive. Nonetheless, I think that authors must clarify some aspects, along with more significant details in the methods. More importantly, the discussion is incomplete, and it is perhaps the section that will demand more work.
  • Abstract and title: The title does not reflect the sample, particularly due to the nationality of the recruited participants. Moreover, the abstract does not provide an introductory sentence. Please, do not add a sentence referring to the pandemic itself; rather, convince your audience in terms of the contribution of studying cortisol levels and its correlates.
  • Essential contents: The abstract is the place to “sell” your paper. To increase visibility, add details of the sample (key demographics) and be generous when presenting the results (such as adding point estimates, 95%CI, effect sizes and so on). Your conclusion is built upon data not presented in the abstract; double-check this, please.
  • Introduction: The first paragraphs focus too heavily on aspects that we are all aware off. Instead, a suggestion would be revamping your rationale around aspects such as behavior and physiology insights that can be obtained during stressful situations, including the lockdown. This apparent simple shift can make a big difference.
  • All the studies about correlates of the pandemic could be summarized by one or two systematic reviews, leaving extra space for the rationale around behavioral and physiological insights.
  • Add a subheading when shifting to the case of medical students that were unable to leave the country. The same is valid for the discussion of cortisol, which could be expanded. Consider a Figure to illustrate the mechanism of cortisol secretion and associated systems.
  • When presenting hypotheses, add references, please.
  • Methods: Please, complete and upload the Strobe statement to properly describe this section. Major elements are missing or, when present, can be more logically place in specific headings and subheadings.
  • Separate procedures from participants and design.
  • Present instruments in separate paragraphs. Reliability indices are missing. Preferably, perform EFA or CFA.
  • Why did you use Fisher’s test and not X2? Is it due to small individuals per cell? This needs to be explained, hence the utility of guidelines such as the Strobe statement. Include achieved power based on your effect size measures (likely n2). Moreover, the mathematical modelling reported in lines 136-160 should be placed under data analysis.
  • Results: Why not begin with a Table with descriptive statistics off all IVs?
  • Section 3.1.1: did you take into account the fluctuations given women’s the well-known effect of the luteal phase? This could be a significant bias that does not impede your analysis but must be described. Strobe statement guidelines suggest sensitive or subgroup analysis that might assist you.
  • The point of introducing a descriptive table (see comment 12) aid in understanding statements like this “A total PSS-10 score can range from 0 to 40, with higher scores indicating higher stress levels. Overall, female participants had significantly higher total PSS-10 scores than male participants”. You need to present the means, SD, and Cohen’s d.
  • I think that this comparison does not make sense “No sex-specific differences were found in response frequency to any of the 10 items (Fisher’s exact test, p > 0.05), suggesting similar response patterns across sexes (Figure A1).”
  • Page 8, lines 290-306: Consider using just the PSS subfactors for more robust comparisons (helplessness and  self-efficacy). The same could be used in Table 1 (PSS total score and subfactors). Results depicted in Table 1 must explicit which period of data collection they refer to.
  • I am not sure if the KFF Tracking Health Poll is useful for your paper. I’ll get rid of it, or move it to supplementary material. Table 3 I would also put as a supplementary material and keep in the text only statistics.
  • I am wondering why correlations were not presented. You have excellent data and at different time points.
  • Discussion: Please, restructure to a) restate your goals; b) discuss the key findings; and c) move to implications and limitations. Subheadings may assist you in better conveying this section. As it stands, it is quite difficult to follow and there is constant shit between topics.
  • Remove descriptions of the measures and place them in the appropriate sections, namely: methods. Example: “The PSS-10 is a widely used psychological instrument for measuring perceived stress. Developed by Sheldon Cohen and colleagues in 1983 [37], it evaluates the extent to which individuals perceive their lives as being unpredictable, beyond their control, and excessively demanding”. The Strobe statement will be of immense help for you.
  • Overall, the discussion is confusing and repeats either the results or descriptions of constructs and instruments. If authors consider what I have said in comment 4, there is plenty of material that can be better organized and discussed. Unfortunately, as it stands, the discussion does not meet the criteria of scientific rigor.
  • Conclusion might be subject to change if sensitive/subgroup analysis are performed.

Author Response

Dear Reviewer, thank you for your insightful comments and constructive suggestions. We have carefully addressed each point, and you will find our detailed responses in the enclosed document

Reviewer 2 Report

Comments and Suggestions for Authors

see attached file

Comments on the Quality of English Language

The English is generally understandable, but the manuscript would benefit from a careful language edit. Several sentences are long or unclear, and there are minor grammatical issues and occasional repetitions. A light professional proofreading would improve clarity and readability.

Author Response

Dear Reviewer, thank you for your insightful comments and constructive suggestions. We have carefully addressed each point, and you will find our detailed responses in the enclosed document.

Round 2

Reviewer 1 Report

Comments and Suggestions for Authors

Great revision. All the points in my review report were addressed.